Progress of research into circular RNAs in urinary neoplasms

Wan Bangbei 1
Liu Bo 2
Lv Cai lvcai815@163.com 1
1 Department of Urology, Central South University Xiangya School of Medicine Affiliated Haikou Hospital , Haikou , China
2 Laboratory of Developmental Cell Biology and Disease, School of Ophthalmology and Optometry and Eye Hospital, Wenzhou Medical University , Wenzhou , China
Lambrughi Matteo
Electronic publication date: 2020 Feb 24
Publication date: 2020
Volume: 8
Electronic Location ID: e8666
Received 2019 Oct 18; Accepted 2020 Jan 30
Copyright: ©2020 Wan et al.
Copyright year: 2020
Copyright holder: Wan et al.
License: This is an open access article distributed under the terms of the Creative Commons Attribution License, which permits unrestricted use, distribution, reproduction and adaptation in any medium and for any purpose provided that it is properly attributed. For attribution, the original author(s), title, publication source (PeerJ) and either DOI or URL of the article must be cited.
License URL: https://creativecommons.org/licenses/by/4.0/

Keywords: circRNA, Bladder cancer, Renal cancer, Prostate cancer

Funding: Natural Science Foundation of Hainan Province 819MS136 The study was supported by the Natural Science Foundation of Hainan Province (No. 819MS136). The funders had no role in study design, data collection and analysis, decision to publish, or preparation of the manuscript.

==============================
Circular RNAs (circRNAs) are a large class of endogenous RNA that form a covalently closed continuous loop without 5′ or 3′ tails and are diffusely expressed in mammalian cells. Through the development of high-throughput sequencing, microarray, and bioinformatics analyses, recent studies have shown that the expression of circRNAs is dysregulated in human tumor tissues and cells, as well as in the blood of patients, and closely correlates with the development of tumors. circRNAs can regulate the progression of tumors through various mechanisms. An increasing number of studies have shown that circRNAs may play critical roles in the early diagnosis, targeted therapy, and prognostic prediction of cancer as biomarkers or therapeutic targets. This review briefly describes the definitions and functions of circRNAs, and the main content includes the most recent progress in research into their function, regulation, and clinical relevance to bladder, renal, and prostate cancers. We also provide some novel ideas regarding the treatment of these diseases.

Introduction

Noncoding RNA (ncRNA) includes the major classes of housekeeping and regulatory ncRNAs. Housekeeping ncRNAs include ribosomal RNA (rRNA), tRNA and small nuclear RNA (snRNA). Regulatory ncRNAs can be classified according to length (<200 or >200 bp), including small noncoding RNA that contain microRNA (miRNA), snRNA, piRNA, siRNA, and long ncRNA (lncRNA). Circular RNAs (circRNAs; >200 bp) are unusual endogenous noncoding RNAs (Bolha, Ravnik-Glavač & Glavač, 2017) that were first reported by Nigro et al. (1991). They were initially considered to be splicing errors with no function (Cocquerelle et al., 1993), but through the recent development of high-throughput sequencing, microarray, and bioinformatics analyses, recent studies have shown that circRNAs may play pivotal roles in the occurrence and progression of disease, including pancreatic cancer (An et al., 2018), breast cancer (Zhao et al., 2018), epilepsy (Gong et al., 2018), and cardiovascular diseases (Wang et al., 2017a; Wang et al., 2017b). The present review mainly highlights the most recent progress in research related to the functions, regulation, and clinical relevance of circRNAs in bladder cancer (BC), renal cell carcinoma (RCC), and prostate cancer (PCa) (Table 1) and provides novel ideas for future investigations and the treatment of these diseases.

Survey methodology

We systematically searched titles in the PubMed database using the term, “circular RNA” combined with “cancer”, “guidelines”, “prostate cancer”, “renal cancer”, “bladder cancer”, “database”, “data”, and “identification”. Publications that were unrelated to human circular RNA were excluded. We did not refine factors such as journal, publishing date, or journal impact factors during our search.

Definition of circRNAs

circRNAs comprise a class of non-coding RNA that have different structures from linear RNAs, such as lncRNA and miRNA, and form covalently closed, continuous stable loop structures without 3′ or 5′ ends. Back-splicing or lariat-circulating events might play vital roles in the formation of integral circRNA. Most circRNAs are exons that stem from intergenic regions, but a few originate from intergenic, intronic, antisense or untranslated regions (Fu et al., 2018). According to their biogenesis, circRNAs can be categorized as exonic (ecircRNAs), circular intronic (ciRNAs), retained-intron or exon-intron (EIciRNAs), and intergenic (Bolha, Ravnik-Glavač & Glavač, 2017). Most circRNAs are located in the cytoplasm; however, some also reside in subcellular compartments such as exosomes and mitochondria (Tian et al., 2018; Szabo & Salzman, 2016; Shao et al., 2017; Chen et al., 2017; Zhang et al., 2019) (Fig. 1A). Intron length, exon length, respective sequences, and RNA-binding proteins (RBPs) can affect the formation of circRNAs (Fu et al., 2018).

Functions of circRNAs

circRNAs can function as miRNA sponges

circRNAs harbor miRNA binding sites within their sequences and can directly bind to miRNA, inhibit combinations between miRNA and 3′-untranslated regions (3′-UTR) of target genes, and abolish the effects of miRNA on gene expression (Chen & Duan, 2018; Li et al., 2018; Bian et al., 2018). This is the primary mechanism involved in the circRNA regulation of gene expression. circ-SFMBT2 can serve as a sponge for miR-182-5p to regulate CREB1 mRNA expression and promote the proliferation of gastric cancer cells (Sun et al., 2018). circ-0000523 inhibits colorectal cancer, and its overexpression attenuates the proliferation of cancer cells and induces apoptosis. circ-0000523 can interact with miR-31 and inhibit the Wnt/ β-catenin signaling pathway, and thus, it can regulate the growth of colorectal cancer cells (Jin et al., 2018). circ-NEK6 functions as an oncogenic circRNA in thyroid cancer as it is significantly up-regulated and can promote the proliferation and invasion of cancer cells. The results of mechanical trials have suggested that circ-NEK6 increases FZD8 mRNA expression, activates the wnt signaling pathway by interacting with miR-370-3p, which subsequently promotes the progression of thyroid cancer (Chen et al., 2018). In addition, Wang et al. (2018a), Wang et al. (2018b) and Wang et al. (2018c) found that circ-DOCK1 is distinctly upregulated in oral squamous cell carcinoma (OSCC) cell lines and OSCC tissues, and that circ-DOCK1 downregulation can induce apoptosis in OSCC cell lines. circ-DOCK1 can also increase BIRC3 expression by acting as a sponge for miR-196a-5p, and then participating in the process of OSCC apoptosis.

Translation

circRNA differs from linear RNA. Some circRNAs possess an open reading frame; hence, they might be able to code proteins or peptides. One study has suggested that a circRNA (220 nt) from the rice yellow mottle virus can encode a highly basic, 16-kDa protein (AbouHaidar et al., 2014). One 2017 study found that consensus N6-methyladenosine (m6A) motifs and a single m6A site are sufficient to drive translation initiation, and proved that protein can be translated by circRNAs in human cells. Further studies have indicated that the m6A-driven translation of circRNAs requires the initiation factor eIF4G2 and m6A reader YTHDF3. Methyltransferase METTL3/14 can enhance, whereas demethylase FTO can inhibit m6A-driven translation of circRNAs (Yang et al., 2017). Moreover, others have found that circRNAs with internal ribosome entry site elements (IRES) or prokaryotic ribosome-binding sites can encode peptides (Perriman & Ares Jr, 1998; Chen & Sarnow, 1995).

circRNAs as gene transcribers and expression regulators

Some circRNAs that circulate with introns “retained” between exons are called intron-containing circRNAs, such as ciRNA and EIciRNA, which mainly reside in the nucleus, interact with U1 snRNP, and enhance the transcription of their parental genes in a cis-acting manner (Li et al., 2015). In addition, some circRNAs, such as circ-ankrd52 and circ-sirt7, can accumulate at the transcriptional sites of host genes, interact with RNA polymerase II (pol II) complexes, and then regulate the transcription of the parental genes (Chen, 2016). In addition, the synthesis of back-spliced circRNAs competes with pre-mRNA splicing and leads to lower levels of linear mRNA that result in regulated gene expression (Ashwal-Fluss et al., 2014).

Protein binding

circRNAs can bind, store, sequester, and interact with proteins to regulate the expression and translation of genes. Many circRNAs interact with RNA binding proteins (RBPs). For example, the circRNA, circ-Mbl, and its flanking introns have conserved muscleblind (MBL) protein binding sites that firmly and specifically bind MBL. Modulating MBL levels significantly affects the biosynthesis of circ-Mbl (Ashwal-Fluss et al., 2014). Du et al. (2016) reported that high levels of circ-Foxo3 expression in non-cancer cells are associated with cell cycle progression. More concretely, aberrant circ-Foxo3 expression represses cell cycle progression by binding to the cell cycle proteins cyclin-dependent kinase 2 (CDK2) and cyclin-dependent kinase inhibitor 1 (P21), which results in the formation of a ternary complex and subsequently blocks cell cycle progression. Chen et al. (2019) found high levels of circ-AGO2 and human antigen R (HuR) expression and associated them with poor outcomes among patients with gastric cancer. circ-AGO2 promotes the growth, invasion, and metastasis of cancer cells in vitro and in vivo. The mechanisms involve physical interactions between circ-AGO2 and HuR protein to promote its activation and enrichment on the 3′-untranslated region of target genes. This in turn results in decreased circ-AGO2 binding and the inhibition of gene silencing modulated by circ-AGO2/miRNA that correlates with cancer progression.

circRNAs and bladder cancer (BC)

BC is the most prevalent among malignant tumors of the urinary system. The incidence of BC is the highest among urogenital tumors in China, and it is second only to PCa in Western countries. The incidence of BC in males is 3 to 4-fold higher than that in females (Bray et al., 2018), and it increases annually, with the incidence reaching a peak at ages between 50 and 70 years. Hence, biomarkers are urgently needed to diagnose and treat BC. circ-RNA shows considerable potential to serve as such biomarkers. Many studies have indicated that circRNAs are involved in the occurrence and progression of BC through multiple mechanisms.

Inhibitor activities

circRNAs function as sponges for miRNA to upregulate target gene expression and inhibit the progression of BC (Fig. 1B). circ-HIPK3 is a circ-RNA derived from Exon2 of the HIPK3 gene (Zheng et al., 2016). Silencing circ-HIPK3 can significantly inhibit the growth of human cells by directly binding to and inhibiting miR-124 activity (Zheng et al., 2016). Dysregulated circ-HIPK3 expression correlates with the occurrence and progression of human diseases (Cao et al., 2018; Yu, Chen & Jiang, 2018) and circ-HIPK3 functions differ among diseases. Li et al. (2017a) and Li et al. (2017b) found that circ-HIPK3 is significantly decreased in BC tissues and cell lines, while it negatively correlates with BC grade, invasion, and lymph node metastasis. The overexpression of circ-HIPK3 effectively inhibits the migration, invasion, and angiogenesis of BC cells in vitro and suppresses BC growth and metastasis in vivo. Mechanistic studies have shown that circ-HIPK3 harbors two vital binding sites for miR-558 and can abundantly sponge miR-558 to suppress the expression of HPSE, VEGF, MMP9, thereby suppressing the progression of BC. Therefore, circ-HIPK3 might be a novel biomarker for treating BC. Chi et al. (2019) reported that circ-000285 derived from HIPK3 inhibits BC. Analysis using quantitative real-time polymerase chain reaction (qRT-PCR) has shown that circ-000285 is downregulated in BC tissues, cell lines, and sera derived from patients with BC. In addition, the expression of circ-000285 is decreased almost three-fold among patients who are resistant to cisplatin compared with those who are cisplatin-sensitive. Additionally, circ-0000285 is associated with tumor size, differentiation, lymph node metastasis, distant metastasis, and TNM stage. Collectively, these findings indicate that circ-000285 can serve as a biomarker for BC diagnosis and chemotherapy, as well as a prognostic predictor. circ-BCRC4 is an important inhibitor of BC because its overexpression attenuates cancer cell proliferation and induces apoptosis. circ-BCRC4 also interacts with miR-101 to downregulate EZH2 expression and induce cancer cell apoptosis (Li et al., 2017a; Li et al., 2017b).

Table 1 Summary of circRNAs in bladder, renal, and prostate cancers.

Cancer type	circRNA	miRNA	Downstream	Function	Reference	
Bladder cancer	circ-HIPK3	miR-558	HPSE, VEGF, MMP9	Inhibits migration, invasion, and angiogenesis of cancer cells in vitro, as well as tumor growth and metastasis in vivo.	Li et al. (2017a) and Li et al. (2017b)	
	circ-BCRC4	miR-101	EZH2	Attenuates proliferation of cancer cells and induces apoptosis.	Li et al. (2017a) and Li et al. (2017b)	
	circ-ITCH	miR-17 miR-224	P21, PTEN	Induces G1 cell cycle arrest and apoptosis, inhibiting BC cell proliferation, invasion, and migration in vitro, as well as tumor growth in vivo.	Yang et al. (2018)	
	circ-BCRC-3	miR-182-5p	P27	Attenuates proliferation, cell cycle progression of BC cell in vitro; inhibits tumor growth in vivo.	Xie et al. (2018)	
	circ-UBXN7	miR-1247-3p	B4GALT3	Inhibits cancer cell proliferation, invasion, and migration in vitro; represses tumor growth in vivo.	Liu et al. (2018a) and Liu et al. (2018b)	
	circ-FNDC3B	miR-1178-3p	G3BP2	Inhibits cancer cell proliferation, migration, invasion, tumor growth, and lymphatic metastasis in vitro and in vivo.	Liu et al. (2018a) and Liu et al. (2018b)	
	circ-TCF25	miR-103a-3p and miR-107	CDK6	Promotes cancer cell proliferation, migration in vitro, tumor growth, and metastasis in vivo.	Zhong, Lv & Chen (2016)	
	circ-BPTF	miR-31-5p	RAB27A	Promotes cancer cell invasion, migration in vitro, and tumor growth in vivo.	Bi et al. (2018)	
	circ-VANGL1	miR-605-3p	VANGL1	Promotes cancer cell proliferation, migration, metastasis, cell cycle progression in vitro, and tumor growth in vivo.	Zeng et al. (2019)	
	circ-0058063	miR-145-5p	CDK6	Promotes cancer cell proliferation, migration, metastasis, and cell cycle progression; inhibits apoptosis in vitro.	Sun et al. (2019)	
	circ-MYLK	miR-29a	CD31, S100A4, ZO-1, VEGFA, Ras, p-Raf-1, p-MEK1/2, p-ERK1/2	Promotes cancer cell proliferation, invasion, and migration, inhibiting apoptosis in vitro; facilitates tumor growth, EMT, and metastasis in vivo.	Zhong et al. (2017)	
	circ-0000144	miR-217	RUNX2	Promotes cancer cell proliferation, invasion, and migration in vitro, as well as tumor growth in vivo.	Huang et al. (2018)	
	circ-UVRAG	miR-223	FGFR2	Promotes cancer cell proliferation, migration, and metastasis in vitro, as well as tumor growth in vivo.	Yang et al. (2019)	
Renal cell cancer	circ-0001451	–	–	Suppresses cancer cell proliferation; promotes apoptosis in vitro.	Wang et al. (2018a) and Wang et al. (2018b)	
	circ-ATP2B1	miR-204-3p	FN1	Suppresses cancer cell invasion in vitro and tumor growth in vivo.	Han et al. (2018)	
	circ-HIAT1	miR-195-5p
miR-29a-3p
miR-29c-3p	CDC42	Suppresses cancer cell invasion and migration in vitro, as well as tumor growth in vivo.	Wang et al. (2017a) and Wang et al. (2017b)	
Prostate cancer	circ-MYLK	miR-29a	–	Promotes cancer cell proliferation, invasion, and migration; decreases apoptosis in vitro.	Dai et al. (2018)	
	circ-SMARCA5	–	–	Promotes cancer cell proliferation and cell cycle progression in vitro.	Kong et al. (2017)	

Figure 1 Locations and functions of circRNAs.

(A) Primarily, circRNAs are located in the cytoplasm, but some reside in subcellular compartments such as exosomes and mitochondria. (B) Function of circRNAs. circRNAs serve as sponges that directly bind miRNA or protein to regulate gene expression and activate signaling pathways, thus participating in the development of bladder, renal, and prostate cancers.

circRNAs also function as sponges of miRNAs to upregulate target gene expression and inhibit BC progression (Fig. 1B). For example, recent studies suggest that circ-ITCH expression is significantly downregulated in several cancer cell lines and tumor types (Luo, Gao & Sun, 2018; Wang et al., 2018a; Wang et al., 2018b; Wang et al., 2018c). Enforced circ-ITCH expression can significantly suppress BC cell proliferation, migration, invasion, and metastasis, as well as induce apoptosis and G1/S cell cycle arrest in vitro and in vivo. Mechanistically, circ-ITCH enhances the expression of miR-17, miR-224 target gene p21 and PTEN by “sponging” miR-17 and miR-224, which suppresses the aggressive biological behavior of BC (Yang et al., 2018). Xie et al. (2018) found that low levels of circ-BCRC-3 are expressed in BC tissues and cell lines, and that enhanced circ-BCRC-3 expression can attenuate BC cell proliferation in vitro and in vivo. Mechanistically, circ-BCRC-3 enhances the expression of miR-182-5p target gene cyclin-dependent kinase inhibitor 1B (p27) by “sponging” miR-182-5p, which attenuates the aggressive biological behaviors of BC. Methyl jasmonate (MJ), which has anticancer effects, can suppress BC progression by increasing circ-BCRC-3 expression. Hence, circ-BCRC-3 functions as a tumor inhibitor that suppresses BC progression through the miR-182-5p/p27 axis, which would be a novel target for BC therapy. circ-UBXN7 is also a novel supressor of BC, and miR-1247-3p is a carcinogenic factor, of which B4GALT3 is a target. Recent findings suggest that circ-UBXN7 is significantly downregulated in BC tissues compared with matched non-tumor tissues, and that circ-UBXN7 expression is related to the pathological stage, grade, and prognosis of BC. Others have shown that circ-UBXN7 directly binds miR-1247-3p, and can reverse the oncogenic effects induced by miR-1247-3p. Mechanistically, circ-UBXN7 can function as a competitive endogenous RNA (ceRNA) of miR-1247-3p that enhances B4GALT3 expression, consequently inhibiting BC cell viability and invasion. Consequently, the circ-UBXN7-miR-1247-3p-B4GALT3 regulatory network should be a potential target for treating BC (Liu et al., 2018a; Liu et al., 2018b). Liu et al. (2018a) and Liu et al. (2018b) reported that circ-FNDC3B also functions as a tumor suppressor in BC. circ-FNDC3B is a highly stable and cytoplasmic circRNA derived from exons 5 and 6 of the FNDC3B gene locus that is expressed at low levels in BC cell lines and tissues, and associated with pathological T stage, grade, lymphatic invasion, and the overall survival rates of patients. Functional experiments have shown that overexpressed circ-FNDC3B significantly inhibits the proliferation, migration, invasion, growth and lymphatic metastasis of tumors both in vitro and in vivo. Mechanical experiments have shown that miR-1178-3p is a carcinogenic factor that targets the 5′ UTR of G3BP2. circ-FNDC3B can function as a miR-1178-3p sponge that directly binds to miR-1178-3p to regulate the expression of G3BP2, inhibit the SRC/FAK signaling pathway, and suppress the development of BC.

Promoter activities

circRNAs are not only inhibitors but also promoters of BC. circRNA can elevate target gene expression by combining with miRNA and thus promote BC progression (Fig. 1B). Zhong, Lv & Chen (2016) used microarrays to determine circRNA expression profiles in BC tissues and found that circ-TCF25 is upregulated. Furthermore, circ-TCF25 overexpression can downregulate miR-103a-3p and miR-107, increase CDK6 expression and promote BC cell proliferation and migration in vitro and in vivo. Hence, circ-TCF25 might be a novel promising marker for BC. circ-BPTF is a novel circRNA derived from BPTF exons, and it is significantly upregulated in BC, compared with adjacent normal tissues and cell lines. Patients who express more circ-BPTF have higher BC tumor grades and poorer prognoses. Functionally, circ-BPTF knockdown reduces tumor progression in vitro and in vivo. Mechanistically, mimics of miR-31-5p, which is a target miRNA of circ-BPTF, can partially reverse the effects of circ-BPTF. Furthermore, RAB27A is a target of miR-31-5p, and circ-BPTF weakens the anti-oncogenic effects of miR-31-5p and consequently promotes RAB27A expression. Therefore, attenuating circ-BPTF expression might be a promising direction to pursue as a strategy for treating BC (Bi et al., 2018). Zeng et al. (2019) found that levels of circ-VANGL1 BC are higher in BC than in adjacent normal tissues. Functionally, silencing circ-VANGL1 significantly attenuates BC cell proliferation, cell cycle, migration, and invasion in vitro and suppresses tumor growth in vivo. Mechanistically, circ-VANGL1 directly binds to miR-605-3p. The interaction between circ-VANGL1 and miR-605-3p inhibits miR-605-3p expression, by elevating that of VANGL1 that facilitates BC progression. Sun et al. (2019) reported that circ-0058063 is upregulated in BC tissues, and that silencing circ-0058063 attenuates the migration capacity of cancer cells, increases cell apoptosis rates, and arrests cells at the G0/G1 phase in vitro, thereby suppressing tumor growth in vivo. Mechanistically, circ-0058063 can negatively modulate miR-145-5p as a sponge, and elevate CDK6 expression to promote BC progression.

Microarray analysis has shown that the circRNA, circ-MYLK, is significantly upregulated in BC. Levels of circ-MYLK are associated with the progression of BC stage and grade. Functionally, circ-MYLK overexpression noticeably enhances BC cell invasive and migratory capabilities. Moreover, upregulating circ-MYLK can promote the epithelial-mesenchymal transition (EMT) of BC. Mechanistically, circ-MYLK overexpression enhances the expression of CD31, S100A4, VEGFA, snail, N-cadherin, vimentin, Ras, p-RAF-1, p-MEK1/2, pERK1/2, and suppresses that of E-cadherin and ZO-1. circ-MYLK functions as a ceRNA for miR-29a, thus activating VEGFA/VEGFR2 and the downstream Ras/ERK signaling pathway to exert these effects (Zhong et al., 2017). Huang et al. (2018) identified circ-0000144 as a novel oncogene in BC, because its expression is significantly upregulated in BC, compared with adjacent non-tumor tissues, and high levels of circ-0000144 expression are associated with a poor prognosis. Functionally, circ-0000144 knockdown attenuates BC cell proliferation and invasion in vitro, and reduces tumor volumes in vivo. Mechanistically, circ-0000144 is a sponge for miR-217, and RUNX2 is a target of miR-217. circ-0000144 enhances RUNX2 expression by repressing miR-217 and thus exerts oncogenic activities in BC.

circ-UVRAG is derived and cyclized by some exons from the UVRAG gene. A recent study indicated that circ-UVRAG is up-regulated in BC lines. Functionally, silencing circ-UVRAG suppresses tumor formation and metastasis. Mechanistically, miR-223 is a downstream binding target of circ-UVRAG, and miR-223 can target FGFR2. Downregulated circ-UVRAG promotes miR-223 expression, which weakens FGFR2 expression in BC. Therefore, circ-UVRAG might serve as a good diagnostic biomarker of BC, and contribute to targeted BC therapy (Yang et al., 2019). Overall, the above evidence suggests that circRNAs have the potential to serve as novel biomarkers for the diagnosis, therapy, and prognostic prediction of BC.

At present, these studies have mainly investigated conventional functions (ceRNA mechanisms) of circRNAs in BC. Unconventional functions of circRNAs, such as translation and protein binding, should be further explored. Defining these functions will contribute to a deeper understanding of the mechanisms underlying the involvement of circRNAs in cancer development.

circRNA and renal cancer

Renal cancer also is a common urological tumor that was responsible for over 403,000 new diagnoses in 2018 and an estimated 175,000 deaths worldwide (Bray et al., 2018). Therefore, identifying novel biomarkers for the diagnosis and treatment of renal cancer might reduce its mortality. circRNAs play major roles in RCC (Xiong, Zhang & Song, 2019) (Fig. 1B).

Clear cell RCC (ccRCC) is a common pathological type of renal carcinoma. Wang et al. (2018a), Wang et al. (2018b) and Wang et al. (2018c) showed that circ-0001451 is significantly down-regulated in ccRCC tissues, and that it correlates with the clinicopathological features (including clinical stage, tumor stage, lymph node, and metastasis) and overall survival of patients with ccRCC. Functional experiments have confirmed that circ-0001451 knockdown promotes the proliferation of ccRCC cell lines and inhibits apoptosis in vitro, indicating its potential as novel diagnostic biomarker of ccRCC and a potential target for ccRCC treatment. However, the mechanism of ccRCC suppression requires further investigation. Estrogen receptor beta (ER β) is a carcinogenic factor in ccRCC. Increased ER β expression is associated with a more advanced stage and a poor prognosis for patients with ccRCC. Mechanistically, ER β represses circ-ATP2B1 expression by inhibiting its host gene ATP2B1 and directly binding to the ERE (-1765 to -1760 nt) on the 5′-promoter of ATP2B1. Moreover, ER β-circ-ATP2B1 might function by interacting with miR-204-3p, resulting in decreased miR-204-3p and increased FN1 mRNA expression that promotes ccRCC progression (Han et al., 2018). Wang et al. (2017a) and Wang et al. (2017b) reported that androgen receptors (AR) interact with circ-HIAT1 and promote ccRCC development. Overall survival of is worse among patients with ccRCC who express low, compared with high levels of circ-HIAT1. Functionally, AR promote ccRCC cell proliferation, migration and invasion. Mechanistically, AR suppress circ-HIAT1 expression by inhibiting its host gene (HIAT1) at the transcriptional level and directly binding to the ARE on the HIAT1 promoter, regulating miR-195-5p/29a-3p/29c-3p expression, and thereby enhancing CDC42 mRNA expression to promote ccRCC progression. In summary, the above evidence suggests that circRNAs could serve as prognostic biomarkers and therapeutic targets for ccRCC. Many circRNAs exist in RCC, but only a few have been investigated. The manifold functions and mechanisms of circRNAs in RCC should be further explored in depth. In addition, conventional functions such as ceRNA mechanisms and unconventional functions such as translation and protein binding warrant further investigation.

circRNA and prostate cancer (PCa)

PCa is frequently diagnosed in elderly men. One report has indicated that about 1.3 million new diagnoses of PCa and 359,000 associated deaths will occur worldwide during 2018 (Bray et al., 2018), and that morbidity and mortality rates will continue to increase. circRNAs play vital roles in PCa development (Fig. 1B). The relative mRNA expression of circRNAs can be modulated by upstream or downstream factors to regulate PCa progression. Notably, miR-145 is a tumor suppressor in cancer, He et al. (2018) found using microarray analysis that miR-145 overexpression promotes the expression of circRNA-101981, circRNA-008068, and circRNA-406557, and suppresses that of circRNA-101996 and circRNA-091420 in PCa LNCaP cells. Therefore, circRNAs can be regarded as regulatory factors in PCa development. circMYLK is not only a cancerogenic factor in BC but also in PCa. Dai et al. (2018) found significantly upregulated circ-MYLK expression in PCa tissues and cancer cell lines. Functionally, upregulated circRNA-MYLK significantly promotes tumor cell proliferation, invasion, and migration while decreasing cell apoptosis, whereas circ-MYLK knockdown reverses these processes. Mechanistically, circ-MYLK promotes PCa cell proliferation, colony formation, invasion, and migration by downregulating miR-29a expression. Kong et al. (2017) found high expression levels of circ-SMARCA5 in PCa samples compared with matched noncancerous prostate tissues, indicating that circ-SMARCA5 could be an oncogenic circRNA. Functional experiments have shown that silencing circ-SMARCA5 in PCa cells suppresses proliferation, increases the amount of cells in G1 phase, decreases that in S phase, and elevates apoptosis rates. However, the mechanisms of circ-SMARCA5 in PCa have not been investigated in detail. Presently, studies into the roles of circRNA in PCa are scarce, and further detailed investigations into the mechanisms of circRNAs are required because circRNAs have the potential to serve as therapeutic and prognostic targets of PCa.

Conclusion

circRNA expression markedly differs between cancer and healthy cells. circRNA function as oncogenic factors or tumor suppressors that participate in cancer occurrence and progression through a multitude of mechanisms. Dysregulated circRNA expression is inextricably associated with cancer development. The expression of circRNAs correlates with clinical stage, tumor stage, lymph node metastasis, and the prognosis of patients with cancer. circRNAs functionally influence the proliferation, invasion, migration, cell cycle progression, apoptosis, and drug resistance of cancer cells. circRNAs mechanistically act as sponges of miRNAs to indirectly regulate target gene expression. Moreover, circRNAs can function upstream as proteins that regulate target gene expression. circRNAs participate in cancer development through all these mechanisms. Therefore, measures are needed to decrease cancerogenic circRNAs and/or enhance the expression of circRNA tumor-suppressors as novel targets of cancer therapy.

Additional Information and Declarations

Competing Interests

Author Contributions

Data Availability

The authors declare there are no competing interests.

Bangbei Wan conceived and designed the experiments, analyzed the data, prepared figures and/or tables, authored or reviewed drafts of the paper, and approved the final draft.

Bo Liu analyzed the data, prepared figures and/or tables, authored or reviewed drafts of the paper, and approved the final draft.

Cai Lv conceived and designed the experiments, authored or reviewed drafts of the paper, and approved the final draft.

The following information was supplied regarding data availability:

This is a literature review and there is no raw data.

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
