# Peer review of "Progress of research into circular RNAs in urinary neoplasms"

_PeerJ, doi:10.7717/peerj.8666_

## Round 0.1 · original submission · Minor Revisions

Your manuscript has now been reviewed and the reviewers' comments are appended below. I agree with them in finding overall your manuscript relevant and interesting however, they have raised points that need to be addressed by a minor revision.

We therefore invite you to revise and resubmit your manuscript, taking into account the points raised and carefully answering them.

Please submit a revised paper and a point-by-point response to the referees.

We look forward to hearing from you soon.
Best regards

Reviewer 1 ·

Basic reporting

No comments

Experimental design

No comments

Validity of the findings

No comments

Additional comments

In this manuscript, Wan and colleagues reviewed the current knowledge of circRNAs in bladder cancer, renal cancer, and prostate cancer. The authors also discussed circRNAs as therapeutic targets for the treatment of these diseases. Overall, the review is novel and interesting. However, I have a few minor comments to improve the manuscript.
Comments:
1. Figure legends need to be rewritten for a better understanding of the figures.
2. Figure 1; The circRNA function has been very well discussed in many reviews. Please remove the circRNA function from this figure.
3. Figure 2; All transcript starts with Exon 1, please edit the figure accordingly and put another exon at the end. Additionally, the vast majority of circRNAs start and end in the middle exons. The circRNA function panel in Fig 1 can be placed in the place of the middle panel in Fig 2.

·

Basic reporting

In this review, author summarized the role of circ-RNA in urinary neoplasms. The article has good logicality and the narration is fluent.

Experimental design

1.In this article, author choosed bladder cancer, renal cancer, and prostate cancer. Are the the include all types of urinary neoplasms or occupy most of urinary neoplasms? Please explain the relationship between these three kinds of tumors and urethral tumors to make convenient for readers without medical background.

Validity of the findings

2.There are some inconsistent abbreviations such as “CircRNA or circRNA”, Anticancer or anticancer? (line 238), and mir or miR? (line 285).
3.Plesas properly introduce the naming rules of CicrRNA as I feel the name of circRNA is confused.
4.In line 200, there is a instance about circHIPK3, the author side “Silencing circHIPK3 can significantly inhibit the growth of human cells ” and in line 206 the author side “Overexpression of circHIPK3 effectively inhibited migration, 207 invasion, and angiogenesis of bladder cancer cells in vitro and suppressed bladder cancer growth 208 and metastasis in vivo”. it seems there is contradiction between this sentences. If there is different results about circHIPK3 in different study? Please interpret and discuss it.
5.The sentece “Recent studies showed that dysregulation of circHIPK3 expression correlated with occurrence and progression of human disease (Cao et al., 2018;Yu et al., 2018)” line202-203 should be put ahead to make the narrative more logical.
6.Please add the “q-PCR” abbreviation behind “Quantitative Real-Time Polymerase Chain Reaction” (line 213). Most readers are more familiar with q-PCR.
7.Line 214, author has mentioned “circRNA-000285 was down-regulated in bladder cancer tissues, cell lines and serum of bladder cancer patients”, the said “ it was also significantly down-regulated in bladder cancer serum samples” again (line 215). It was duplicate of serum.
8.Line 252, “CircFNDC3B was.....” if there should be “is”?
9.Line 197, in general understanding, miRNA can inhibit target gene expression, here the author side “CircRNAs function as sponges of miRNAs to down-regulate target gene expression and inhibit progression of bladder cancer (Fig2)”. So how circRNA sponges miRNA and further down-regulated target gene expression?
10.There are some sentence shuld be split, for instance in line 243, the author said “A recent study suggested that circUBXN7 was significantly down-regulated in BC tissues compared with matched non-tumor tissues, and the expression of circUBXN7 was related to the pathological stage, grade and poor prognosis of bladder cancer positively, further studies found that circUBXN7 could directly bind to miR-1247-3p, and reversed the oncogenic effects induced by miR-1247-3p”. the sentence is too long, and further studies...should be another new sentence. Similar problems occur many times.

Additional comments

11.If the each circRNA and instances could be summerized in table, it will convenient for readers to remember and summarize.
12.Last but not least, in general, most part of the text is simply narrate, there is few of valuable discussion and author's own thoughts. Pleases discuss further, for example, but not limited to what is the appropriate concentration of circRNA treatment in examples involved, which is a problem of concern of readers, and what should be paid attention to in circRNA studing? And what are the biggest advantages or obstacles in circRNA studing field in author's point of view?

---

## Round 0.2 · accepted · Accept

The last version of the manuscript has been checked. In light of the reviewer's advice, I am delighted to say that we are happy to publish the revised version of the manuscript in PeerJ.